# 2.5D Cascaded Semantic Segmentation for Kidney Tumor Cyst

Zhiwei Chen[1], Hanqiang Liu[2]

School of Science, Shaanxi Normal University, China
chenzhiwei@snnu.edu.cn

**Abstract.** Accurate segmentation of kidney tumours can help doctors diagnose the disease. In this work, we described a multi-stage 2.5D semantic segmentation networks to automatically segment kidney and tumor and cyst in computed tomography (CT) images. First, the kidney is pre-segmented by the first stage network ResSENormUnet; then, the kidney and the tumor and cyst are fine-segmented by the second stage network DenseTransUnet, and finally, a post-processing operation based on a 3D connected region is used for the removal of false-positive segmentation results. We evaluate this approach in the KiTS21 challenge, which shows promising performance.

**Keywords:** Multi-stage, Kidney and tumor and cyst segmentation, Deep Learning

## 1 Introduction

Kidney cancer is the cancer of the genitourinary system with the highest mortality rate [1]. There are many ways to treat kidney tumors, and segmentation is only one of them. If the results of segmentation are valid, it will be helpful for subsequent tumor detection and treatment. Currently, clinical practice mainly relies on manual segmentation of the kidney and tumor, but manual segmentation brings problems such as time-consuming and laborious, and also causes inconsistent segmentation results due to differences in the subjective perceptions of physicians, which makes preoperative planning difficult, thus the need for automated segmentation is becoming more and more urgent.

In the work, motivated by the [2], we developed a two-stage neural network to locate and segment the kidney, tumor and cyst from 3D volumetric CT images. It consists of two main stages: one is rough kidney localization and the other is accurate kidney, tumor and cyst segmentation. In the first stage, only the kidney (with tumor and cyst) is segmented, and the segmentation results are stored to obtain the region of interest (ROI) and leave out the outside pixels to mitigate class imbalance and reduce memory consumption; in the second stage, we train a fine segmentation network based on the cropped kidney region obtained in the first stage. Finally, the predicted mask of the target region is transformed into a volume of the original size.

## 2 Methods

Since the kidneys make up only a small portion of the entire CT image. Each case may include non-kidney region, its segmentation is easily misled by unrelated tissues. In addition, direct segmentation of the kidney, tumor and cyst can cause difficulties in segmentation due to differences in tumor and cyst sizes and blurred boundaries between the two. This class imbalance leads to extremely difficult identification and segmentation, so the model is trained in multi-stage. Therefore, first, we train a ResSENormUNet to get a coarse segmentation of the kidney with the region of interest (ROI) in the volume. The extracted kidney then into DenseTransUNet for kidney, tumor, and cyst segmentation. As shown in Figure 1. Finally, the results are post-processed.

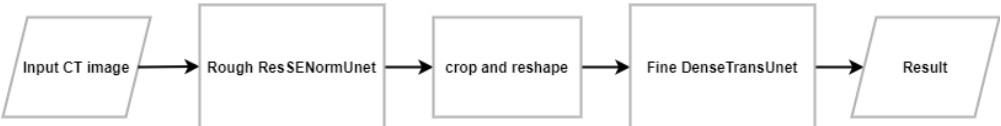

**Fig.1. The pipeline of our method**

## 2.1 Training and Validation Data

There are 300 and 100 abdominal CT scans for training and testing in the KiTS21 Challenge dataset, respectively. Our submission made use of the official KiTS21 training set alone. We split the given 300 training CT volumes into 240 for training and 60 for validation, and evaluate the segmentation accuracy using the Dice score. For the training set, we input the original image in the first stage of the network, and in the second stage we will crop the original image and the labels directly according to the labels of the original image, and then input the second stage of the network for training. For the validation set, we input the original image for prediction in the first stage, crop the prediction result and then input it to the second stage network for further refinement.

## 2.2 Preprocessing

Firstly, we truncated the image intensity values of all images to the range of [-79, 304] HU to remove the irrelevant details. The choice of HU boundary values is referred to [6]. Then, truncated intensity values are normalized into the range of [0, 1] using a min-max normalization. Normalization over entire image in stage 1 benefits ROI extraction and normalization over solely ROI enhances learning targets so as to facilitate model learning [3]. Since the medical image acquisition is difficult, the amount of data is small and time-consuming to label, it is necessary to choose data enhancement for the data, which can not only add more equivalent data on the original data, but also improve the generalization ability of the model. In this paper, the data augmentation methods include horizontal flipping, random brightness contrast, random gamma, grid distortion, and flat reduction rotation, etc.

## 2.3 Proposed Method

### 2.3.1 Kidney Localization

Stage 1 of our model uses a 2.5D approach to find the ROI position of the kidney in the volume. Therefore, we merge kidney, tumor, and cyst of the target to one class. 2.5D can take more contextual information between slices into account compared to 2D, extracting more adequate features while reducing memory pressure, improving training speed compared to the 3D approach, and producing more accurate results than the 2D approach. The network structure of this stage is shown in Figure 2, and the model is a UNet-like convolutional neural network. The model input is a stack 9 slice of adjacent axial, providing large image content in the axial plane. The model output is a segmentation map corresponding to the center slice of the stack. In the encoder, the first convolutional kernel size after the input is $7 \times 7$ to increase the perceptual field of the model without incurring significant computational overhead. The remaining convolutional kernels are all of size $3 \times 3$. The step sizes are all 2. The Rectified Linear Unit (ReLU) is used as the nonlinear activation function. Both SE Norm [4] and multiscale supervision are added. SE Norm can effectively improve the performance of the model. The SE Norm combines Squeeze-and-Excitation (SE) blocks with normalization. Similar to Instance Normalization, SE Norm layer first normalizes all channels of each example in a batch using the mean and standard deviation. Secondly, apply global average pooling (GAP) to squeeze each channel into a single descriptor. Then, two fully connected (FC) layers aim at capturing non-linear cross-channel dependencies. Besides, we apply deep supervision to enhance the discriminative ability of medium-level features. In each decoder level, we use a convolution and an upsampling to get the same spatial size as the original image and calculate the loss using the masks from all level. The advantage is that each upsampling will be as similar as possible to the target.

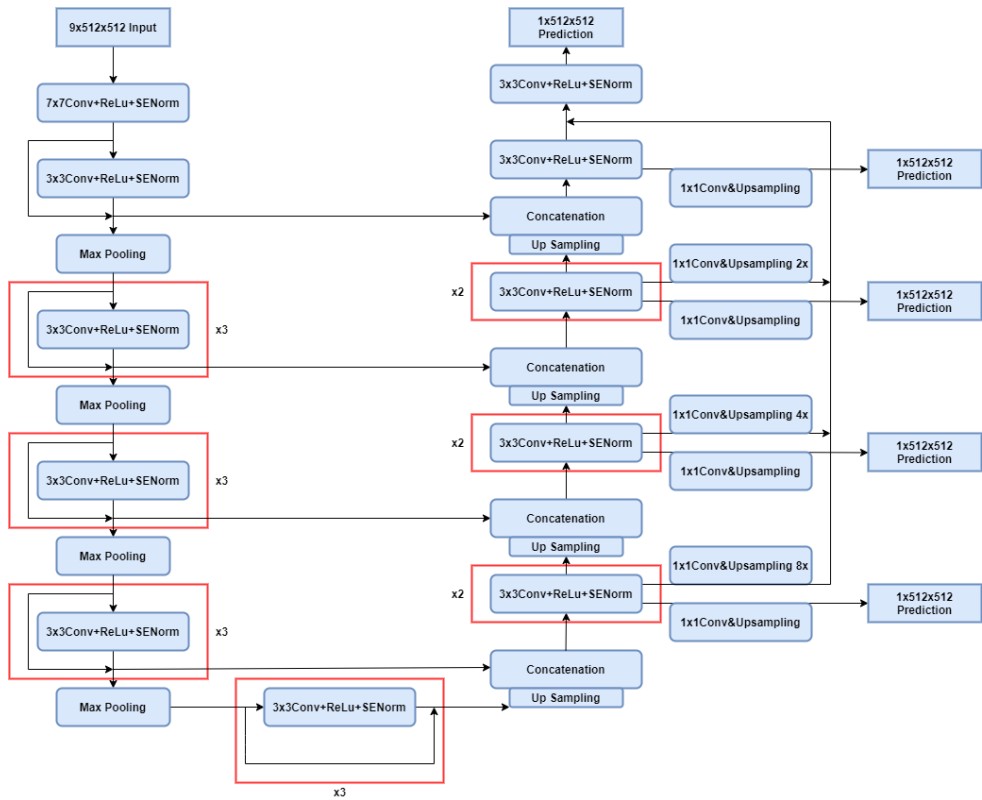

**Fig.2. Architecture for kidney coarse segmentation**

### 2.3.2 Kidney Tumor Cyst Segmentation

The stage 2 aims at further segmentation of the kidney, tumors and cysts, and this stage uses the region of interest obtained in the previous stage as network input. The network structure of this phase is shown in Table 1, with densenet161 as the backbone network. Because adding a densely connected network is proven to enhance feature propagation, encourage feature reuse, and improve the network's ability to identify features, while mitigating the problem of gradient disappearance. In addition, adding Transformer to the model. Transformer is designed to model long-range dependencies in sequence-to-sequence tasks and capture the relations between arbitrary positions in the sequence [5]. It is powerful in modeling global context. Same as the stage 1, we also added deep supervision in DenseTransUnet.

**Table 1.** Architectures of the DenseTransUNet. The symbol k means kernel size, s means stride, p means padding and ch means output channels. "( ) $\times d$" means this block is repeated for d times.

| Name | Ops | Feature map ($h \times w$) |
|---|---|---|
| input | - | $512 \times 512$ |
| convolution 1 | $BN + ReLU + conv(k = 7, s = 2, p = 3, ch = 96)$ | $256 \times 256$ |
| pooling | $max\ pool(k = 3, s = 2, p = 1)$ | $128 \times 128$ |
| dense block 1 | $\begin{pmatrix} BN + ReLU + conv(k = 1, ch = 192) \\ BN + ReLU + conv(k = 3, p = 1, ch = 48) \end{pmatrix} \times 6$ | $128 \times 128$ |
| transition layer 1 | $BN + ReLU + conv(k = 1) + average\ pool(k = 2)$ | $64 \times 64$ |
| dense block 2 | $\begin{pmatrix} BN + ReLU + conv(k = 1, ch = 192) \\ BN + ReLU + conv(k = 3, p = 1, ch = 48) \end{pmatrix} \times 12$ | $64 \times 64$ |

| transition layer 2 | BN + ReLU + conv(k = 1) + average pool(k = 2) | $32 \times 32$ |
|---|---|---|
| dense block 3 | $\begin{pmatrix} \text{BN + ReLU + conv(k = 1, ch = 192)} \\ \text{BN + ReLU + conv(k = 3, p = 1, ch = 48)} \end{pmatrix} \times 36$ | $32 \times 32$ |
| transition layer 3 | BN + ReLU + conv(k = 1) + average pool(k = 2) | $16 \times 16$ |
| dense block 4 | $\begin{pmatrix} \text{BN + ReLU + conv(k = 1, ch = 192)} \\ \text{BN + ReLU + conv(k = 3, p = 1, ch = 48)} \end{pmatrix} \times 24$ | $16 \times 16$ |
| Transformer layer | conv(k = 3, ch = 2208, s = 1, p = 1) + reshape 
 +Multi-Head Attention(MHA) block 
 +Feed Forward Network(FFN) + reshape | $1 \times 256$ |
| upsampling layer 1 | transposed conv(k = 3, s = 2, p = 1) 
 +skip connection(dense block 3) 
 +conv(k = 3, p = 1, ch = 768) + BN + ReLU | $32 \times 32$ |
| upsampling layer 2 | transposed conv(k = 3, s = 2, p = 1) 
 +skip connection(dense block 2) 
 +conv(k = 3, p = 1, ch = 384) + BN + ReLU | $64 \times 64$ |
| upsampling layer 3 | transposed conv(k = 3, s = 2, p = 1) 
 +skip connection(dense block 1) 
 +conv(k = 3, p = 1, ch = 96) + BN + ReLU | $128 \times 128$ |
| upsampling layer 4 | transposed conv(k = 3, s = 2, p = 1) 
 +skip connection(convolution 1) 
 +conv(k = 3, p = 1, ch = 96) + BN + ReLU | $256 \times 256$ |
| upsampling layer 5 | transposed conv(k = 3, s = 2, p = 1) 
 +conv(k = 3, p = 1, ch = 96) + BN + ReLU | $512 \times 512$ |
| convolution 2 | k = 3, p = 1, ch = 4 | $512 \times 512$ |

### 2.3.3 Postprocessing

A post-processing method based on three-dimensional connectivity domain analysis was used to calculate the area of the region consisting of each detected marker object, leaving only the portion of the region area larger than the threshold as the maximum connected region of the kidney with or without cancerous tissue. Since the tumor will connect with the kidney and given by the prior knowledge that no more than two kidneys exist in the abdomen. Therefore, first, we merge the kidneys and tumor of the segmentation result and ignore the background. If this component is smaller than the second largest component multiplied by 0.8, we remove it. Second, we perform another post-processing based on the connected region only for the tumor. We will remove if this component is smaller than the largest component multiplied by 0.4.

### 2.3.4 Loss and Optimization

All networks are trained with stochastic gradient descent and a batch size of 16. The unweighted sum of the Generalized Dice Loss and the Focal Loss is utilized to train the model. Use Adam optimizer with initial learning rate of 1e-4 and multiplied by 0.1 when loss is not decrement in 5 epochs. Each model was trained for 100 epochs.

## 3 Results

We use the Dice coefficient, which is widely used in medical image segmentation, to quantitatively evaluate the accuracy of the model. An example of our prediction results is depicted in Fig. 3.

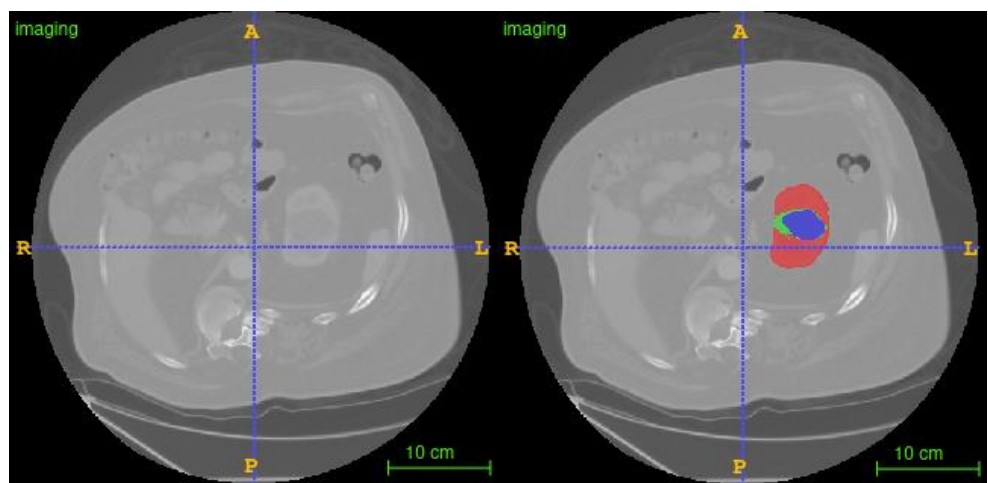

**Fig. 3.** An example of prediction results of case 256. The kidney is shown in red, the tumor in green, and the cyst in blue.

Table 2 shows the results of our model on the validation dataset. Training was done on Nvidia GeForce RTX 3090 GPU(single GPU training). All networks were implemented with the PyTorch framework. It took about 5 days for training the model in the first stage and about 2 days for it in the second stage.

**Table 2.** The experimental results in validation data using our method.

|  | Kidney | Tumor | Cyst |
|---|---|---|---|
| Average Dice on Validation Dataset | 0.9430 | 0.7779 | 0.7099 |

## 4 Discussion and Conclusion

In this paper, we propose a segmentation method based on a multi-stage stepwise refinement approach for the segmentation of kidney, tumor and cyst in abdominal enhanced CT images. A 2.5D approach is used for data input in network training to preserve certain contextual semantic information while relieving memory pressure. In addition, this paper adopts a post-processing method based on the 3D connected domain to remove the false positive regions in the segmentation results and further improve the segmentation accuracy. There are issues in this paper that need further study and the network and methods can be improved for smaller kidneys, tumor and cyst segmentation.

## Acknowledgements

This work is supported by the National Natural Science Foundation of China (Grant Nos. 62071379, 62071378 and 61571361), the Natural Science Basic Research Plan in Shaanxi Province of China (Grant Nos. 2021JM-461 and 2020JM-299), the Fundamental Research Funds for the Central Universities (Grant No. GK201903092), and New Star Team of Xi'an University of Posts & Telecommunications (Grant No. xyt2016-01).

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
