# OpenReview forum: "2.5D Cascaded Semantic Segmentation for Kidney Tumor Cyst"
_MICCAI.org/2021/Challenge/KiTS — Submitted to KiTS21 Challenge_

### Official Review · Reviewer_cr8t · 2021-08-30

**Rating:** 7

**Review:**

The authors present a two-stage 2.5D approach to the problem that first roughly segments the kidneys and then filters the ROIs with connected component analysis. Finally, a second network is used to perform fine-grained segmentation within the identified regions of interest. The paper does an adequate job of covering the necessary details, but one crucial detail that is missing is the method that was used to aggregate the different segmentation instances into a composite that could be used for training and validation. Most teams used majority voting, is that the case here as well? If so you should state this explicitly in your methods.

---

### Official Review · Reviewer_TLWv · 2021-08-30

**Rating:** 7

**Review:**

### Overall

- Segmentation is not exactly *necessary* for kidney cancer treatment, but it can be helpful

### Introduction

- Looks good - just see comment above

### Methods

- How did you choose your HU bounds?

### Results

- Please make sure to add your official results once they are known

### Discussion and Conclusion

- This section should be numbered 4 instead of 3
- You say "vesicle" when I think you mean cyst

---

### Decision · Program_Chairs · 2021-08-30

**Decision:**

Minor Revisions

**Comment:**

Please address the reviewer comments and resubmit